# Hansen Solubility Parameter Analysis on Dispersion of Oleylamine-Capped Silver Nanoinks and their Sintered Film Morphology

**DOI:** 10.3390/nano12122004

**Published:** 2022-06-10

**Authors:** Satoshi Saita, Shin-ichi Takeda, Hideya Kawasaki

**Affiliations:** 1Department of Chemistry and Materials Engineering, Faculty of Chemistry, Materials and Bioengineering, Kansai University, Osaka 564-8680, Japan; k854423@kansai-u.ac.jp; 2Takeda Colloid Techno-Consulting Co., Ltd., Osaka 564-0051, Japan

**Keywords:** Hansen solubility parameter, dispersibility, Ag nanoparticles, oleylamine, nanoink, printed electronics

## Abstract

Optimizing stabilizers and solvents is crucial for obtaining highly dispersed nanoparticle inks. Generally, nonpolar (hydrophobic) ligand-stabilized nanoparticles show superior dispersibility in nonpolar solvents, whereas polar ligand (hydrophilic)-stabilized nanoparticles exhibit high dispersibility in polar solvents. However, these properties are too qualitative to select optimum stabilizers and solvents for stable nanoparticle inks, and researchers often rely on their experiences. This study presents a Hansen solubility parameter (HSP)-based analysis of the dispersibility of oleylamine-capped silver nanoparticle (OAm-Ag NP) inks for optimizing ink preparation. We determined the HSP sphere of the OAm-Ag NPs, defined as the center coordinate, and the interaction radius in 3D HSP space. The solvent’s HSP inside the HSP sphere causes high dispersibility of the OAm-Ag NPs in the solvent. In contrast, the HSPs outside the sphere resulted in low dispersibility in the solvent. Thus, we can quantitatively predict the dispersibility of the OAm-Ag NPs in a given solvent using the HSP approach. Moreover, the HSP sphere method can establish a correlation between the dispersibility of the particles in inks and the sintered film morphology, facilitating electronic application of the nanoparticle inks. The HSP method is also helpful for optimizing stabilizers and solvents for stable nanoparticle inks in printed electronics.

## 1. Introduction

Nanoparticles (NPs) with sizes of <100 nm have been explored for their novel physical, chemical, electronic, magnetic, optoelectronic, and catalytic properties, and have been widely applied in various fields [1,2,3]. Printed electronics can be used to form conductive wires in electronic components, owing to the printing technology [4,5]. Compared to conventional methods (e.g., photolithography and vacuum processes), the printing technology is more straightforward and has a lower environmental impact. Conductive inks contain conductive particles (e.g., metal nanoparticles, conductive polymers, carbon materials and organic/metallic compounds), solvents, and other additives [6,7,8]. Formulating stable conductive inks is needed for producing high-quality printed electronics [9,10,11]. In particular, NPs have high surface energy and tend to aggregate in inks. The low dispersion stability of the nanoparticle ink reduces the performance of electrical products. Therefore, the development of an effective strategy for improving the dispersion stability of nanoparticle inks is of great significance.

NP dispersion is determined by the affinity of the NP surface toward solvents. Ligand (e.g., surfactant and polymers) stabilizers on the particle surface can tailor the interfacial affinity between the NP surface and solvent molecules [12,13]. Thus, the optimization of ligand stabilizers and solvents is crucial for obtaining highly dispersed NP inks [2]. Generally, nonpolar (hydrophobic) ligand-stabilized NPs show superior dispersibility in nonpolar solvents (e.g., toluene and hexane). Meanwhile, polar ligand (hydrophilic)-stabilized NPs exhibit high dispersibility in polar solvents (e.g., water and ethanol). However, these properties are too qualitative for selecting optimum ligand stabilizers and solvents for stable nanoparticle inks; thus, researchers often rely on their experiences.

The Hansen solubility parameter (HSP) was developed to predict liquid miscibility based on the total cohesive energy density measured by evaporation experiments [14,15]. The HSP describes the cohesion properties of a liquid; the dispersion (*δ^*2*^_D_*) and the dipole–dipole (*δ^*2*^_P_*) and hydrogen-bonding interactions (*δ^*2*^_H_*) contribute to the total cohesion energy of a liquid (*δ**^*2*^**_total_*):
*δ^*2*^_total_* = *δ^*2*^_D_* + *δ^*2*^_P_* + *δ^*2*^_H_*(1)

The HSP theory is based on the “like seeks like” principle, with likeness measured by the HSP distance metric. A given solute’s critical condition for solubility is inclusion in a three-dimensional (3D) HSP space, as shown in Figure 1a [14,15]. The center of the sphere is the HSP value of the target solute, and the interaction radius (*R_*0*_*) defines the limits of solute solubility in a given solvent. We determine the “distance” (*R_a_*) between the HSP value of the target solute and that of a solvent in the 3D HSP space. A relative energy difference (RED = *R_a_/R_*0*_*) value lower than 1.0 (i.e., inside the sphere) indicates high solubility of the solute in the solvent. In contrast, an RED value greater than 1.0 (i.e., outside the sphere) indicates low solubility. Based on the HSP approach, we can quantitatively predict solute solubility in a given solvent.

Recently, the HSP sphere has been extended to the prediction of particle dispersion in solvents, including carbon, metal oxides, and metal nanoparticles [16,17,18,19,20,21,22,23,24,25,26]. The particle’s HSP sphere was based on particle dispersibility experiments of various solvents (high/low dispersibility). Similar to solute solubility, an RED value of <1 suggests stronger affinity between the particle surface and the solvent, resulting in high dispersibility, as shown in Figure 1b. Peterson et al. determined the HSP of decanoic acid-capped silver NP (Ag NP)/polymer ink by Ag NP dispersion tests in various solvents [18], and optimized the ink solvent to obtain the desirable optical properties of the Ag/polymer film based on the HSP sphere method. Yamamoto et al. reported the influence of various silane coupling modifications on copper particle dispersion in different solvents [21]. The HSP of the NPs tended to change toward that of the surface-modified silane coupling agent. More recently, Süß et al. proposed the Hansen dispersibility parameters (HDP) in place of HSP because particle dispersions are not thermodynamically stable, in contrast to molecular solubility [19]. Moreover, additional interactions (e.g., van der Waals interactions, electrostatics, and DLVO/non-DLVO) between particles should be considered, instead of being restricted to interactions between molecules only. Thus, for particle dispersion, they applied the HDP sphere method to carbon black and zinc oxide quantum dots [19,23].

Previous reports have mainly focused on the application of the HSP method to particle dispersion. For electronic applications of particle-based inks, it is crucial to understand how the HSP sphere method connects the dispersibility of particles in inks to the sintered film morphology. The aggregation of nanoparticle ink drastically affects the sintered film morphology and film conductivity [27,28,29]. However, few attempts have considered the effectiveness of the HSP sphere method for the sintered film morphology of particle-based inks. Moreover, the oleylamine (OAm) ligand is widely used in the solution-phase synthesis of various metal/metal oxide NPs because of its unique features as a solvent, surfactant, and reducing agent [30,31,32,33,34,35,36,37]. However, the HSP of OAm-capped NPs has not yet been determined. Herein, an HSP-based analysis of the dispersibility of oleylamine-capped Ag NP (OAm-Ag NP) inks and their sintered film morphologies was conducted. Our study focuses on the following: (1) synthesis of OAm-Ag NPs; (2) dispersibility tests of OAm-Ag NPs in various solvents using the analytical centrifugation method to determine the dispersibility of OAm-Ag NPs in a given solvent; (3) formation of a 3D HSP space of OAm-Ag NPs based on the results of the dispersibility tests and prediction of candidate ink solvents (test solvents) using the 3D HSP space; and (4) preparation of OAm-Ag-NP-based inks using the test solvents and observation of the sintered Ag film morphology. Finally, we discussed the effectiveness of the HSP analysis of the dispersibility of the Ag NP inks and their sintered film morphology.

## 2. Materials and Methods

### 2.1. Materials

OAm, decane, oxalic acid (99%), silver nitrate (AgNO_3_, 99.8%), butanol (98.0%), methyl ethyl ketone (99.0%), 1,4-dioxane (>98.0%), pentane (97.0%), tetrachloroethylene (>97.0%), and cyclopentanone (>95.0%) (FUJIFILM Wako Pure Chemical Co., Osaka, Japan); diacetone alcohol (>98.0%), ethylbenzene (>99.0%), ethyl benzoate (>99.0%), butylbenzene (>99.0%), furan (>99.0%), and dimethyl sulfoxide (DMSO) (>99.0%) (Tokyo Chemical Industries, Ltd., Tokyo, Japan); and toluene (>99.5%), acetonitrile (99.5%), acetone (99.5%), cyclohexane (>99.5%), and tetrahydrofuran (>99.5%) (KANTO CHEMICAL Co., Inc., Tokyo, Japan) were used for this study.

### 2.2. Synthesis of OAm-Ag NPs

OAm-Ag NPs were prepared according to a previously reported method [38]. Silver oxalate (Ag_2_C_2_O_4_) was prepared by mixing AgNO_3_ (6.36 g) and oxalic acid (1.68 g) in 50 mL of water. The resultant precipitate was collected by filtration under reduced pressure to obtain silver oxalate. A 1:1 molar ratio of Ag_2_C_2_O_4_ (5.31 g) and OAm (4.67 g) was stirred at 60 °C for 10 min to produce an OAm-Ag complex, which was subsequently heated at 150 °C for 15 min to produce OAm-Ag NPs. The NPs were washed thrice with methanol and the precipitate was dried to obtain purified OAm-Ag NPs.

### 2.3. Characterization of OAm-Ag NPs

The surface ligands of the OAm-Ag NP powders were determined by Fourier transform infrared (FT-IR) spectroscopy (FT/IR-4200, JASCO Corporation, Tokyo, Japan; measurement range: 500–4000 cm^−1^, resolution: 4 cm^−1^, number of scans: 128) using an attenuated total reflection (ATR) instrument (ATR PRO ONE, JASCO Corporation, Tokyo, Japan). The crystal structures of the OAm-Ag NP powders were determined by X-ray diffraction (XRD) analysis (D2 Phaser, Bruker AXS GmbH, Karlsruhe, Germany, 2θ range: 10–80°, Cu-Kα radiation source (λ = 1.5406 Å) at 30 kV accelerating voltage and 10 mA current). Transmission electron microscopy (TEM) of the OAm-Ag NPs was conducted using a microscope (JEOL JEM1400, Tokyo, Japan) operated at 120 kV.

### 2.4. Dispersibility Evaluation of Ag NPs in Solvents by Sedimentation Experiments under Centrifugal Acceleration Field

OAm-Ag NP dispersions (1 wt.% OAm-Ag NP) were prepared by dispersing the OAm-Ag NP powder (0.05 g) in the solvents (5 mL) listed in Table 1. A polyamide cell with an optical path length of 2 mm was filled with 300 mL of the NP dispersion. We evaluated the dispersibility based on the sedimentation behavior of Ag NP dispersions under a centrifugal acceleration field immediately after ultrasonication for 15 min [19]. Large aggregates, which are formed by the low dispersibility of Ag NPs in solvents, resulted in high sedimentation velocity. The sedimentation profile of the OAm-Ag NPs in a given solvent was obtained by detecting the transmission profile at an 870 nm wavelength over the entire sample height using a centrifugal sedimentation analysis instrument (LUMiSizer LS610, LUM GmbH, Berlin, Germany) at 3000 rpm and 25 °C for 180 min. As a measure of stability against aggregation, we determined the sedimentation time (*t**) until the particles reached a defined sedimentation stage.

### 2.5. Estimation of Hansen Solubility Sphere from Particle Dispersibility

The particle dispersibility of the solvents was visualized by plotting the HSP values (δ_d_, δ_p_, and δ_h_) of each solvent in a 3D graph, with the HSP value of the OAm-Ag NPs at the center of the sphere. The HSP sphere of the OAm-Ag NPs was created to contain the data points of the HSP with high dispersibility.

### 2.6. Preparation of OAm-Ag NP Inks and Sintered Ag Films from the Inks

OAm-Ag NP inks (50 wt.% Ag content) were dispersed in the solvents by mixing (ARE-310, Thinky, Tokyo, Japan) at 2000 rpm for 10 min and defoaming at 2200 rpm for 10 min. The OAm-Ag NP inks were applied to polyimide substrates by the drop-casting method, and the coated films were heated at 200 °C for 5 h, producing sintered Ag films. The OAm-Ag NP ink was applied to a polyimide substrate using a spin coater (ACT-300DⅡ, ACTIVE Co., Ltd., Saitama, Japan), and the coated OAm-Ag NP film was heated at 200 °C for 2 h to produce a sintered Ag film with a thickness of 3.0 ± 1.0 μm. The sheet resistance was measured using a four-point probe setup (Loresta AX MCP-T370, Mitsubishi Chemical Analytech Co., Yamamto, Japan).

## 3. Results and Discussion

### 3.1. Characterization of OAm-Ag NPs

The OAm-Ag NPs were synthesized by the thermal decomposition of Ag_2_C_2_O_4_ in the presence of OAm. Figure 2a shows the OAm-Ag NP powder. The TEM image exhibits spherical Ag NPs with a size of 20–30 nm (Figure 2b). The XRD pattern of the OAm-Ag NP powder reveals the crystalline structure of the Ag NPs (Figure 2c). The peaks at 2θ = 38°, 44°, 64°, and 77° were indexed to the (111), (200), (220), and (311) reflections of face-centered cubic Ag NPs. The FT-IR spectrum of the OAm-Ag NP powder (Figure 2d) exhibits peaks at 2915 and 2848 cm^−1^, which could be assigned to the asymmetric and symmetric stretching vibrations of −CH_2_, respectively. The peaks at 3442 and 1617 cm^−1^ can be assigned to the vibrations of the –NH_2_ groups. All bands indicated the presence of OAm molecules on the surface of the Ag NPs.

### 3.2. Dispersibility of OAm-Ag NPs in Different Solvents

The dispersibility of the OAm-Ag NP in different solvents was evaluated on the basis of the sedimentation behavior of the Ag NP dispersion under a centrifugal acceleration field. The presence of large aggregates, due to the low dispersibility of the Ag NPs, accelerated the sedimentation velocity. Figure 3a,b show the sedimentation profiles over time for OAm-Ag NPs in effective (cyclohexane) and non-effective (cyclopentanone) dispersion media, respectively. The optical extinction of the samples was monitored over the cell length of 27 mm from the cell bottom, given as the distance from the center of the rotor (mm); the first and last profiles are marked red and green, respectively. The evolution of the optical extinction profiles was averaged over a range of 115–128 mm, which is denoted as integral extinction. Figure 3c,d show the time profiles of the integral extinction. The faster change in the integral extinction indicates the rapid sedimentation rate of OAm-Ag NPs in the solvent (i.e., low dispersion stability). We estimated the sedimentation time (*t**) when a threshold extinction value of 0.015 was reached (*t**** = 130.7 min for cyclohexane and *t** = 7.0 min for cyclopentanone). Table 1 shows the sedimentation time (*t**) of the OAm-Ag NPs in different solvents used to measure their stability against aggregation.

The solvent viscosity and density differences between the solvent and particles affect the sedimentation time, in addition to the size of the particle aggregates. Thus, we defined the relative sedimentation time (*RST*) value by correcting the sedimentation time (*t**) by the density difference (*ρ_P_* − *ρ_L_*) between the particle (*ρ_P_*) and liquid (*ρ_L_*) and the liquid viscosity (*η_L_*) using Equation (2):
*RST = t* ×* (*ρ_P_*
*−*
*ρ**_L_*)/*η**_L_*(2)

We further normalized each *RST* value using the maximum value of *RST_max_* according to Equation (3) [14,15]:
*RST*_*nor*_ = *RST/RST*_*max*_(3)

Table 1 lists the *RST_nor_* values of the OAm-Ag NPs in various solvents. The *RST_nor_* values are given between zero (i.e., low-dispersibility ranking) and one (i.e., high-dispersibility ranking) for any test solvent, which allows easy ranking of dispersibility.

### 3.3. HSP Determination of OAm-Ag NPs Based on the HSP Sphere

By using the HSPiP software (ver. 5.3), we plotted the HSP values (*δ_D_, δ_P_*, and *δ_H_*) of all solvents examined for the dispersibility of OAm-Ag NPs, as shown in Figure 4a. The blue and red squares represent high- and low-dispersibility-ranked solvents, respectively, while the green sphere represents the OAm-Ag NPs HSP sphere. Based on the data in Table 1, five solvents were found to exhibit high dispersibility (RED < 1). The center coordinate of the HSP sphere was defined as the HSP value (*δ_D*0*_, δ_P*0*_, δ_H*0*_*) of the OAm-Ag NPs, determined to be *δ_D*0*_* = 16.5 MPa^0.5^, *δ_P*0*_* = 2.7 MPa^0.5^, and *δ_H*0*_* = 0.01 MPa^0.5^. The radius (*R_*0*_*) was 4.8 MPa^0.5^. The calculated HSP spheres showed approximately no changes when the top three, four, and five effective solvents with high-dispersibility ranking were used; however, incorporating the sixth effective solvent changed the HSP sphere. Thus, we prepared HSP spheres of the OAm-Ag NPs using the top five solvents.

The distance (*R_a_*) between the HSP value (*δ_D*0*_, δ_P*0*_, δ_H*0*_*) of the OAm-Ag NPs and that of the specific solvent (*δ_D_, δ_P_, δ_H_*) in the 3D HSP space can be used to judge the potential of the NPs to be dispersed in the solvent. Closer HSP values of the solute–solvent pair in the 3D HSP space indicate stronger affinity (i.e., high dispersibility). *R_a_* was calculated using Equation (4):
*R_a_*^2^*=* 4(*δ_D_*
*− δ_D_*_0_)^2^
*+* (*δ_P_*
*− δ_P_*_0_)^2^
*+* (*δ_H_*
*− δ_H_*_0_)^2^(4)

The empirical factor of four is ‘convenient’ to represent the experimental data in the 3D HSP space [14,15].

Herein, we predicted the effective or non-effective solvents for the dispersion of OAm-Ag NPs using the RED values and selected four candidate solvents with different RED values for the preparation of OAm-Ag NP inks, namely, decane (RED = 0.64), butylbenzene (RED = 0.74), ethyl benzoate (RED = 1.64), and DMSO (RED = 2.31).

### 3.4. Verification of Predictive Dispersibilities of OAm-Ag NP Inks Based on the HSP Spheres

We prepared the OAm-Ag NP inks with a high Ag content of 50 wt.% using the four test solvents to verify the prediction from the OAm-Ag NP dispersion with a low Ag (1%) content. We experimentally confirmed that the dispersibility of the OAm-Ag NPs in the test solvents was consistent with the forecast from the HSP model, as shown in the optical images of the OAm-Ag NP inks (Figure 4b). Large aggregates, sedimentation, and phase separation were observed in the inks that used ethyl benzoate and DMSO as solvents. OAm-Ag NPs have high dispersibility in decane. Furthermore, partial agglomeration was observed in the OAm-Ag NP ink prepared from butylbenzene with a higher RED value. Thus, the RED values based on the HSP spheres of OAm-Ag NPs can be used to predict the dispersibility of the NPs in the inks.

### 3.5. Verification of Predictive Sintered Film Morphology from OAm-Ag NP Inks Based on the HSP Spheres

We further explored the impact of the dispersibility of the OAm-Ag NP inks on the morphology of sintered Ag films at 200 °C on polyimide films from the OAm-Ag NP inks prepared using the four test solvents. In addition to the dispersibility of OAm-Ag NPs in the test solvents, the following factors were considered. The rapid evaporation of volatile solvents with low boiling points induces the formation of cracks and pores. Thus, test solvents with close boiling points (*T_b_*) were selected as the ink solvents: decane (*T_b_* = 174 °C), butylbenzene (*T_b_* = 192 °C), ethyl benzoate (*T_b_* = 213 °C), and DMSO (*T_b_* = 189 °C). In addition, the boiling points of the solvents must be close to the sintering temperature (200 °C). The critical surface tension (*γ_c_*, mNm^−^^1^) of the polyimide used as a substrate is 37 mNm^−^^1^ [39], which is the value of the surface tension from the regression equation for cos θ = 1 (contact angle = 0°). To achieve complete wettability of the solvents used for the polyimide substrate, the surface tension (*γ*, mNm^−^^1^) of the test solvents must be lower than the *γ_c_* value of polyimide: decane (*γ* ≈ 24), butylbenzene (*γ* ≈ 29), ethyl benzoate (*γ* ≈ 37) and DMSO (*γ* ≈ 42).

Figure 5a–d show the optical images of the sintered Ag films at 200 °C on a polyimide film from OAm-Ag NP inks. The macroscopic visible fractured Ag film was observed from the ink using DMSO. To observe the microscopic morphology of Ag films, we collected the corresponding SEM images. The cracking of the Ag films decreased when ink solvents with lower RED values were used, as shown in Figure 5e–h. Microscopic cracking occurred in the sintered Ag films from the inks with butylbenzene, ethyl benzoate, and DMSO. However, only minor cracking was observed in the Ag film that used the ink with decane, and the sintered Ag film showed resistivity of 69 μΩcm.

We also found that the crystallite sizes of Ag films increased from 338 Å in DMSO to 396 Å in decane, based on the XRD peaks of the Ag films (Figure 6). Volkman et al. reported that higher aggregation rates in Ag-based inks tend to induce small crystallite sizes, less orientation, and enhanced twinning in particles [40]. Similarly, the aggregated OAm-Ag NP inks suppressed the growth of crystallite sizes during the sintering process. The experimental results confirmed that the ink solvent with the lowest *R*_a_ value (i.e., decane) allows superior dispersibility of the OAm-Ag NPs, develops a film morphology with fewer cracks, and facilitates the growth of crystallite sizes in the sintered Ag films.

The HSP of ligand-capped particles is influenced by various interactions (particle/particle, particle/solvent, particle/ligand, and ligand/solvent). Additionally, desorption of surface ligands from the particles reduces particle dispersibility. Therefore, the HSP value of ligand-capped particles is not always consistent with that of the ligand alone (i.e., the contribution of the ligand/solvent). The comparison of HSP values of OAm-Ag NPs (*δ_D0_* = 16.5 MPa^0.5^, *δ_P0_* = 2.7 MPa^0.5^, *δ_H0_* = 0.01 MPa^0.5^) with those of the OAm ligand (*δ_D_* = 16.2 MPa^0.5^, *δ_P_* = 1.7 MPa^0.5^, *δ_H_* = 3.1 MPa^0.5^) suggests that the contributions from *δ_D_* and *δ_P_* are similar. It is likely that the solvent penetrates the space between the oleyl-alkyl chain of the adsorbed OAm layers on the Ag NPs. The oleyl-alkyl chain has one double bond (C=C) with a “cis” configuration in the middle of the molecule, leading to a shorter extended distance in the solution and bending of the double bond. The loose packing of the OAm molecules on the nanoparticle surface allows the solvent to penetrate the oleyl-alkyl chain layers of Ag NPs (i.e., solvent accommodation effect), resulting in the high dispersibility of NPs [25]. In contrast, the contribution of the *δ_H_* of the OAm ligand, which originates from the amine group (-NH_2_), to the HSP value of OAm-Ag NPs becomes negligible. The amine group of the OAm ligand strongly binds to the surface of the Ag NPs, and the oleyl groups are oriented toward the solution and hide the amine groups of OAm. Therefore, the contribution of the *δ_H_* of the OAm ligand to the HSP value of the OAm-Ag NPs becomes negligible.

## 4. Conclusions

We performed an HSP analysis of OAm-Ag NP dispersibility. We determined the HSP sphere of OAm-Ag NPs after quantifying the dispersibility of OAm-Ag NPs in 15 different solvents. The *R*_a_ between the HSP value of OAm-Ag NPs and that of the specific solvent in the 3D HSP space can be used to judge the potential of the NPs to be dispersed in a solvent. We predicted the efficiency of the solvents for the dispersion of OAm-Ag NPs according to the RED values based on the HSP spheres. We further explored the impact of the RED values on the morphology of sintered Ag films. The experimental results confirmed that the ink solvent with smaller RED values improved the film morphology and growth of crystallite sizes in the sintered Ag films. The current approach demonstrated the applicability of HSPs for predicting the solvent-dependent dispersibility of OAm-Ag NP inks and their sintered Ag film morphology. The HSP approach has also been used to select suitable solvents/nonsolvents for polymers [41]. We expect that our results will facilitate the fabrication of other nanoparticle-based inks with low-molecular ligand/polymer-stabilized nanoparticles and provide predictive insights that can assist in utilizing nanoparticles in electronic applications.

## Figures and Tables

**Figure 1 nanomaterials-12-02004-f001:**
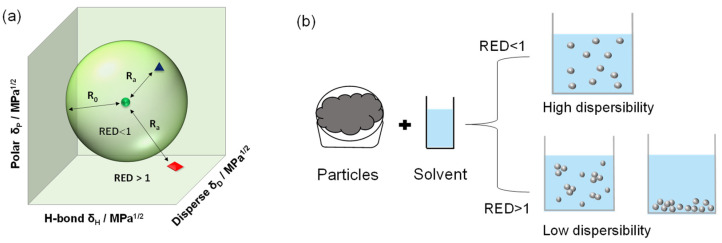
(**a**) The Hansen solubility sphere. The blue triangle indicates high solubility of the solute in the solvent, while the red square indicates low solubility; (**b**) particle dispersibility in solvents. An RED value of <1.0 shows that the particles are evenly distributed throughout the solvent (“High dispersibility”), while an RED value of >1.0 shows particle clumping or settling (“Low dispersibility”).

**Figure 2 nanomaterials-12-02004-f002:**
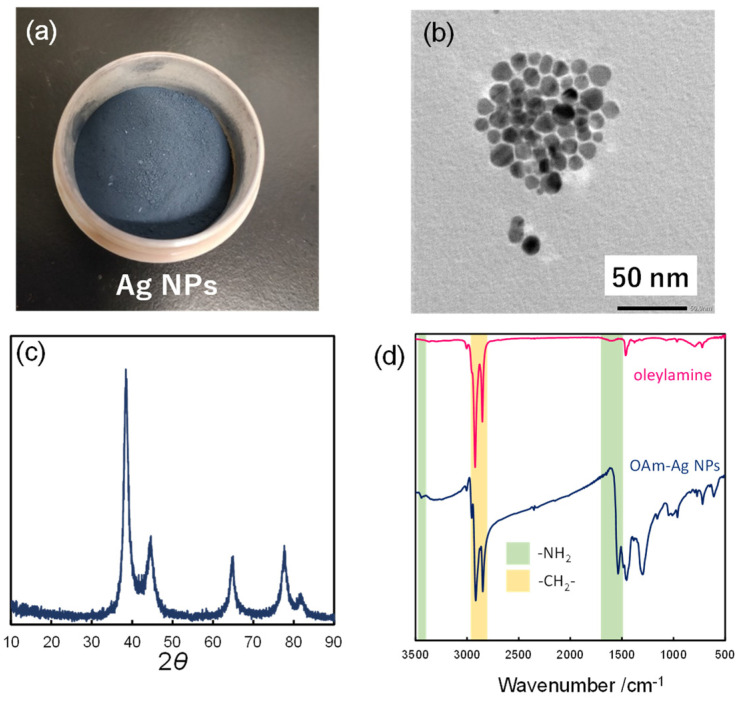
(**a**) OAm-Ag NP powder; (**b**) TEM image of OAm-Ag NPs; (**c**) XRD pattern of OAm-Ag NP powder; and (**d**) FT-IR spectrum of OAm-Ag NPs.

**Figure 3 nanomaterials-12-02004-f003:**
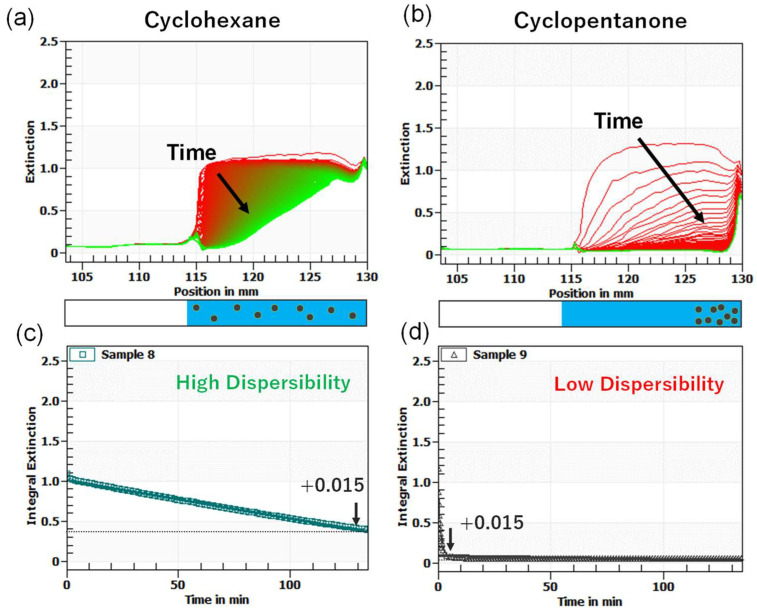
Sedimentation profiles for OAm-Ag NPs in (**a**) cyclohexane and (**b**) cyclopentanone. Integral extinction curves for OAm-Ag NPs in (**c**) cyclohexane and (**d**) cyclopentanone.

**Figure 4 nanomaterials-12-02004-f004:**
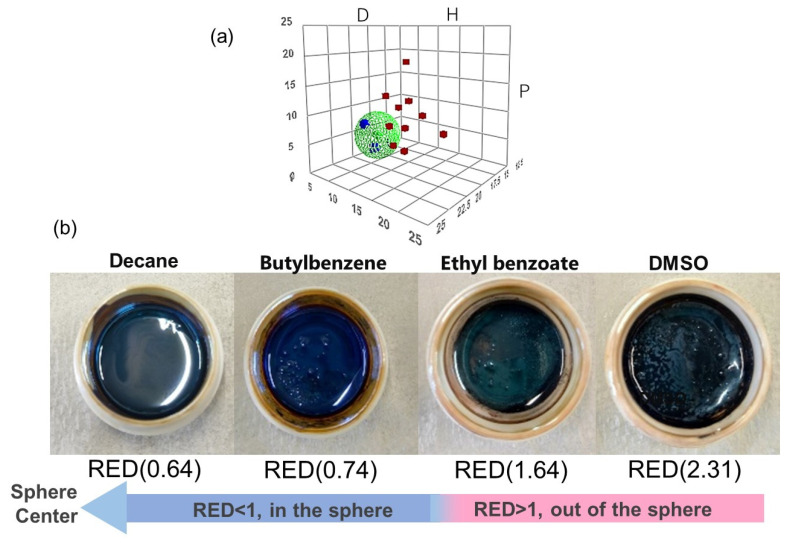
(**a**) HSP sphere of OAm-Ag NPs derived from the HSPiP software. The large green sphere indicates the dispersibility sphere of the OAm-Ag NPs, while the small green sphere in the middle represents the precise coordinates. The red cubes indicate the low-dispersibility-ranked solvents and the blue tetrahedrons indicate the high-dispersibility-ranked solvents and (**b**) OAm-Ag NP inks with decane (RED = 0.64), butylene (RED = 0.74), ethyl benzoate (RED = 1.64), and DMSO (RED = 2.31).

**Figure 5 nanomaterials-12-02004-f005:**
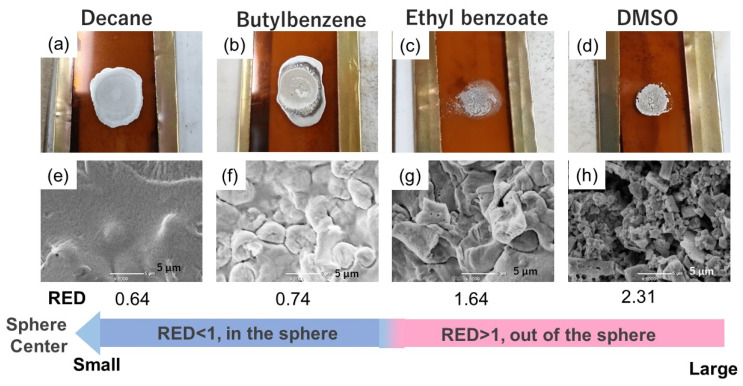
Sintered Ag films at 200 °C using OAm-Ag NP inks with (**a**) decane; (**b**) butylbenzene; (**c**) ethyl benzoate; and (**d**) DMSO. SEM images of sintered Ag films at 200 °C from OAm-Ag NP inks with (**e**) decane; (**f**) butylbenzene; (**g**) ethyl benzoate; and (**h**) DMSO.

**Figure 6 nanomaterials-12-02004-f006:**
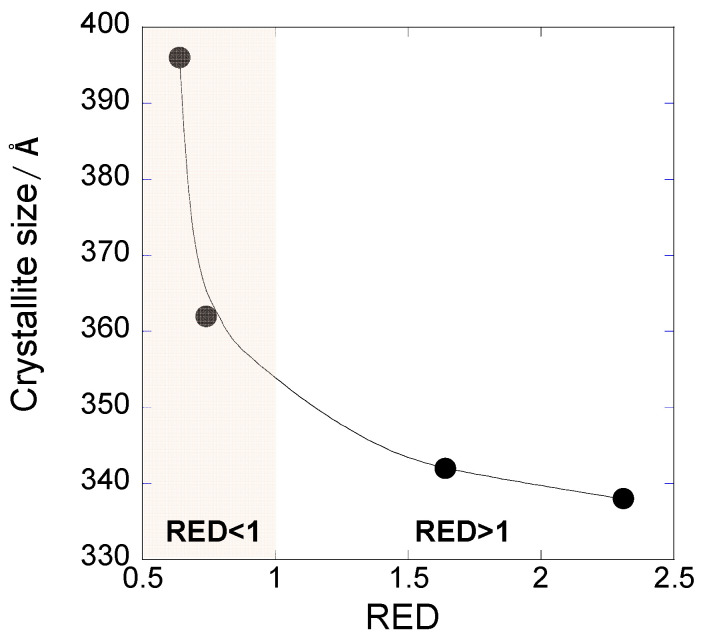
Crystallite sizes of sintered Ag films as a function of the RED values of solvents used for the ink.

**Table 1 nanomaterials-12-02004-t001:** Solubility parameters (MPa^1/2^) and computed RST_nor_ and RED values for solvents used in the study for the determination of HSP.

Solvent	δ_d_ [MPa^1/2^]	δ_p_ [MPa^1/2^]	δ_h_ [MPa^1/2^]	t* [min]	RST_nor_	RED
pentane	14.5	0	0	86.4	1.0	0.99
ethylbenzene	17.8	0.6	1.4	156.0	0.63	0.77
toluene	18	1.4	2	114.8	0.50	0.81
tetrachloroethylene	18.3	5.7	0	160.6	0.43	0.99
cyclohexane	16.8	0	0.2	130.7	0.33	0.58
diacetone alcohol	15.8	8.2	10.8	67.5	0.056	2.5
1-butanol	16	5.7	15.8	48.5	0.047	3.4
ethylbenzoate	17.9	6.2	6	33.1	0.036	1.6
acetone	15.5	10.4	7	3.7	0.025	2.2
acetonitrile	15.3	18	6.1	2.7	0.019	3.5
1,4-dioxane	17.5	1.8	9	7.2	0.013	1.9
cyclopentanone	17.9	11.9	5.2	7.0	0.013	2.3
tetrahydrofuran	16.8	5.7	8	1.5	0.008	1.8
methylethyl ketone	16	9	5.1	0.16	0.0009	1.7
furan	17	1.8	5.3	0.17	0.0003	1.1

## Data Availability

The data presented in this study are available on request from the corresponding author.

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
