# Peer review of "Hansen Solubility Parameter Analysis on Dispersion of Oleylamine-Capped Silver Nanoinks and their Sintered Film Morphology"

_nanomaterials, 2022, doi:10.3390/nano12122004_

Round 1
Reviewer 1 Report
This manuscript describes the prediction of the dispersibility of metal nanoparticles capped with oleylamine using the Hansen solubility parameter approach.
The experiments were systematically performed to demonstrate quantitatively the HSP-based prediction. This approach would be very useful to select solvents for dispersion of organic molecule-capped metal nanoparticles.
Therefore, I recommend the publication of this manuscript in nanomaterials after minor revision.
Additional questions:
Can this prediction be applied in polymer-wrapped metal particle systems?
What about metal nanoparticles capped with hydrophilic end groups?
Author Response
Response to referee comments
Comments from Referee
Reviewer #1: This manuscript describes the prediction of the dispersibility of metal nanoparticles capped with oleylamine using the Hansen solubility parameter approach. The experiments were systematically performed to demonstrate quantitatively the HSP-based prediction. This approach would be very useful to select solvents for dispersion of organic molecule-capped metal nanoparticles. Therefore, I recommend the publication of this manuscript in nanomaterials after minor revision.
Dear Reviewer,
First of all, we would like to thank you for your efforts in referring to our manuscript. We have taken into consideration all of your comments and made the changes. These are listed hereafter.
- Can this prediction be applied in polymer-wrapped metal particle systems?
Response: Thank you for drawing our attention to polymer-wrapped metal particle systems. The Hansen solubility parameter approach has been used to select suitable solvents/nonsolvents for polymers (e.g., Venkatram et al., J. Chem. Inf. Model., 59, 4188(2019)). Thus, we consider that this approach can be applied for polymer-wrapped metal particle systems.
Revision: We added the description of the possibility of the Hansen solubility parameter approach for polymer-wrapped metal particle systems [Page 10, line 335].
- What about metal nanoparticles capped with hydrophilic end groups?
Response: Thank you for the important suggestion. This situation would depend on the chemical species of hydrophilic end groups such as amine, carboxylate, thiol, and hydroxyl groups, but it is interesting to question on the relationship between the chemical structure of ligands and the Hansen solubility parameter approach for the dispersion. In future research, we hope to be able to do such experiments.

Reviewer 2 Report
The work by Saito et al. detail HSP as a parameter to predict solvent stability for nano-particle suspensions and its impact on sintered film morphology. HSP is a well known parameter that is used for discussing stability of colloidal suspensions (J. Electrochem. Soc. 165 F264). The work here is a good description of employment of this parameter to nanoparticle suspensions and provides a good reference for further work. The reviewer recommends its publication.
Small comments:
1. Can the authors briefly comment on other methods of dictating suspension stability - Hildebrand parameter, DLVO/xDLVO models?
Author Response
Comments from Referee
Reviewer #2: The work by Saito et al. details HSP as a parameter to predict solvent stability for nano-particle suspensions and its impact on sintered film morphology. HSP is a well known parameter that is used for discussing stability of colloidal suspensions (J. Electrochem. Soc. 165 F264). The work here is a good description of employment of this parameter to nanoparticle suspensions and provides a good reference for further work. The reviewer recommends its publication.
Dear Reviewer,
First of all, we would like to thank you for your efforts in referring to our manuscript. We have taken into consideration of your comment and made the changes. These are listed hereafter.
1.Can the authors briefly comment on other methods of dictating suspension stability - Hildebrand parameter, DLVO/xDLVO models
Response and Revision: Thank you for the important suggestion. We added the comments on DLVO/xDLVO models in the introduction part [Page 3, line 88].
